# Elucidating the Rhizosphere Associated Bacteria for Environmental Sustainability

**Blessing Chidinma Nwachukwu, Ayansina Segun Ayangbenro** and **Olubukola Oluranti Babalola** *

Food Security and Safety Niche, Faculty of Natural and Agricultural Sciences, North-West University, Private Mail Bag X2046, Mmabatho 2735, South Africa
* Correspondence: olubukola.babalola@nwu.ac.za; Tel.: +27-(0)-183-892-568

**Abstract:** The abundance of nutrient accumulation in rhizosphere soils has placed the rhizosphere as an "epicenter" of bacterial concentrations. Nonetheless, over the years, little attention has been given to bacterial inoculants and soil-like substrates. The reason is that many farmers and experiments have focused on chemical fertilizers as an approach to improve plant growth and yield. Therefore, we focused on assessing the application of rhizosphere soil and its associated bacteria for biotechnological applications. This review has been structured into major subunits: rhizosphere soil as a treasure trove for bacterial community concentration, biodegradation of lignocellulose for biofuel production, rhizosphere soil and its bacteria as soil amendments, and the role of rhizosphere soil and its bacteria for bioremediation and biofiltration. Hence, the efficient use of rhizosphere soil and its bacteria in an environmentally friendly way can contribute to healthy and sustainable environments.

**Keywords:** bioadsorbent; biofuel; bioremediation; plant-microorganism interactions; soil amendment; sustainable development goals; soil health





## 1. Introduction

The United Nations (UN) have released blueprints for seventeen Sustainable Development Goals (SDGs) to achieve a better and more sustainable life for people globally by 2030. The second goal (Zero hunger) is adopted to end hunger and achieve food security by 2030, ensuring that everyone has sufficient food. This goal seeks sustainable solutions to end hunger in all its forms and to achieve food security. The aim is to ensure that everyone has enough good-quality food to lead a healthy life [1]. Therefore, to achieve this goal, there is a need for better access to food and widespread sustainable agriculture promotion. This entails improving the productivity and income of small-scale farmers by promoting equal access to land, technology and markets, sustainable food production systems and resilient agricultural practices. It also requires increased investments through international cooperation to bolster the productive capacity of agriculture, especially in developing countries [1,2].

Over a decade, the fight against hunger globally has progressively improved, and the proportion of the undernourished population decreased from 15% (2000–2002) to 11% (2014–2016) [3]. Nonetheless, approximately 800 million people do not have constant access to quality and safe food. If prevailing conditions persist, the zero hunger objectives will not be achieved by 2030 [4]. The incessant hunger is no longer the problem of food availability. Still, many countries failed to attain the Millennium Development Goals (MDGs) against hunger and human-induced environmental degradation due to advanced food insecurity, especially in sub-Saharan Africa and southern Asia [2]. Hence, there is a need for novel biotechnological applications of beneficial plant growth-promoting bacteria to improve soil fertility and plant health and, as a result, produce sufficient healthy quality food without having any negative impacts on the environment [1].

In agro-based industry, plant root-associated bacteria have a beneficial effect on the growth and yield of crops and forest trees [5,6]. Plant growth-promoting rhizobacteria

(PGPB) are consortia of bacterial species that colonize the plant root region (rhizosphere), impacting plant growth and health advantageously. The PGPB are agricultural bioresources that stimulate plant growth and productivity. They also incite plants' resistance to different phytopathogens in a wide variety of crops including vegetables, fruits, and some trees [7].

The diversity of bacterial species in the rhizosphere has been used as a biological indicator to estimate soil quality and fertility because they play a critical role in nitrogen fixation, hormone production, and nutrient distribution [8]. Similarly, they have contributed to the production and oxidation of methane and acetone, and have resulted in the enhancement of the soil pH, water composition, organic carbon content, and porosity [9,10].

The rhizosphere soils serve as an exclusive natural niche, which houses myriads of bacterial species and their compositions differ with plant species. The most predominant root-associated bacterial community found in the rhizosphere soil are Betaproteobacteria (e.g., *Burkholderia*), Bacteroidetes, Alphaproteobacteria (such as *Rhizobia*), Gammaproteobacteria (like *Pseudomonas*), and Firmicutes (e.g., *Bacillus*) [11]. The rhizosphere soil is composed of a high abundance of bacterial population compared to the bulk soil. These bacteria from the rhizosphere soil can be harnessed and used in an ecofriendly approach as promising biotechnology for the production of antimicrobials, and can serve as biocontrol, bioremediation, and biofertilization agents, thereby improving soil health, soil fertility and crop yield, and ensuring environmental sustainability [12].

Environmental sustainability acknowledges the importance of advancing and controlling the biological and physical systems that bolster both the short- and long-term value of all forms of life on earth without jeopardizing the diversity and well-being of natural ecosystems [13]. By virtue of the ecological services rendered by rhizobacteria, Ambrosini, et al. [14] have recommended further research on the factors that aid in the maintenance of the rhizosphere bacterial community and promote practices that advance rhizosphere conservation and protection. Despite the critical role played by rhizobacteria in redressing soil fertility and environmental sustainability, there still remains the need for further understanding of the mechanisms through which rhizobacteria perform their ecological roles and how such roles can be exploited for environmental sustainability. Therefore, critical discussion on the diversity of bacteria in the rhizosphere soils and their role in lignocellulose degradation, biofuel production, biofiltration, and bioremediation, as well as the possibility of achieving soil amendment, was provided.

## 2. The Rhizosphere Soil as a Treasure Trove for Bacterial Community Concentration

The rhizosphere is known as the region of the soil that surrounds the root where biological, physical and chemical properties of the soil are modulated by plant processes. The rhizosphere is a hotspot of plant-bacteria interplay within the soil environments [15]. It is colonized by diverse bacterial communities, which are functionally and structurally controlled by soil type and texture, environmental factors and plants [16]. Studies have revealed that the plant root exudates and other rhizodeposits lure beneficial bacteria to the rhizosphere, although uninvited ones are also enticed [1]. The host plant induces selection pressure on the development of the rhizosphere microbiome, which favors and attracts a specific plant microbiota due to variations in the composition of the root exudate [17].

Odelade and Babalola [18] stated that there is a higher bacterial biomass in the rhizosphere soil compared to the bulk (rootless) soil, which is as a result of increased availability of substrates for bacterial growth through root exudates, resulting in greater population density and community structure in the root region that may be different from those in the bulk soil. This was supported by a study by De Luna, et al. [19], who stated that the bacterial cell population in 1g of rhizosphere soils is typically $10^8$–$10^{12}$, and they surpassed that of the bulk soil, which is due to various root exudates and rhizodepositions in the root region. This high bacterial density in the rhizosphere soils has been ascribed to the high level of available substrate and humidity [20].

Reports have suggested that root exudates and various rhizodeposits perform key roles in the richness and diversity of bacterial communities in the rhizosphere. Root exu-

dates have the most diversified nutritional composition compared to other rhizodeposits. They are also versatile in composition and influenced by the plant host and environmental factors [21]. The root exudates attract beneficial bacterial species to the rhizosphere of plants, while some unwanted bacterial species are also lured [7]. The constituents of root exudates vary between plant species and cultivars, which leads to variation in the rhizosphere bacterial community. These variations can be manipulated to create specific selective effects on the rhizosphere microbiome [22].

Dennis et al. [23] stated that root exudates carry out a limited role in controlling the bacterial communities in the rhizosphere compared to the other rhizodeposits (volatile compounds, mucilages, slough-off root cells, and lysates) due to large variations in exudate composition and dynamics shown by various studies. Collective evidence indicates that bacterial communities are oftentimes distinct from similar plant cultivars and from bulk soil, but not always. The authors classified these plants as having a delicate rhizospheric effect [24,25]. The mechanisms by which hosts winnow the ambient community to form their microbial communities are not fully understood, although plant functional traits, such as the cuticle composition, may be responsible [15]. However, shaping and establishment of the rhizosphere microbiome is a selective and dynamic process that involves several mechanisms such as signal recognition, chemotaxis, biofilm formation and antibiosis [21].

Root exudation includes the secretion of enzymes, oxygen and water, ion, mucilage and diverse carbon-containing metabolites. The plant root system produces various metabolites, while the root tips secrete most of the root exudates, which are low molecular weight organic substances (such as amino acids, amides, organic acids, sugars, enzymes, phenolic acids and coumarin), high-molecular-weight compounds (such as proteins and mucilages) and other substances, including sterols that attract bacteria to the rhizosphere [26,27]. However, the components of the exudates vary in the amount released, molecular weight, and biochemical functions. These exudates act as attraction signals that influence the ability of bacteria to colonize the roots. To proliferate and be established in the rhizosphere, the organisms must be able to use root exudates, colonize the root or rhizosphere effectively and be able to compete with other organisms [23]. Rhizobacteria locate plant roots through cues exuded from the roots and root exudates, which stimulate Plant Growth-Promoting Rhizobacteria PGPR chemotaxis on root surfaces. Root exudates can also stimulate flagella motility in some rhizobacteria [28]. These traits are essential for the colonization of the rhizosphere.

The impact of plant roots was examined on rhizosphere and bacterial communities, and it was deduced that root length, biomass, density, volume, and surface area create distinct ecological niches for some bacterial species to improve advantageous interactions in the rhizosphere [29]. It has been established that since the root tips make the initial connection with the bulk soil, the bacterial communities and rhizodeposits are notable in maintaining the rhizosphere [4,30].

Despite variations in the dynamics and composition of root exudates, a subset of the bacterial population is designated as the core rhizosphere microbiome, which are ubiquitous across plant species and environment [17]. The microbiota uses the root exudates as a source of energy, and the common genera in the rhizosphere include *Burkholderia, Bacillus, Microbacterium, Azospirillum, Serratia, Pseudomonas, Erwinia, Aeromonas, Mesorhizobium, Rahnella, Acinetobacter, Enterobacter* and *Acinetobacter* [31,32]. Conventionally, bacterial species in the rhizosphere were isolated and identified using the traditional or culture-based method for isolating and classifying microorganisms. This method's main inadequacy is that it cannot identify the entire microorganism in a sample, making approximately 99% of the microorganism unknown [6]. Thus, only a few bacterial populations has been identified from the rhizosphere soils using conventional techniques, but not until the advent of next-generation sequencing techniques [33]. High-throughput sequencing has made the identification of most rhizobacteria possible and also lends credence to their functional role in the rhizosphere (Table 1).

Bacterial communities from the rhizosphere have been implicated in synthesizing extracellular hydrolytic enzymes responsible for biodegradation into the soil. Therefore, they are viewed as the leading force manipulating the terrestrial ecosystems. The abundance of nutrients in the rhizosphere not only contributes to plant growth and development but also maintains the beneficial soil bacterial community inhabiting the rhizosphere soil [1]. Some studies conducted on rhizosphere soils have reported the presence of beneficial bacterial communities essential for biotechnological applications (Table 1). Methanotrophic bacteria capable of producing methane from $NH^{4+}$ were identified from rice paddy rhizosphere soil [34].

**Table 1.** Bacteria present in rhizosphere soil and the techniques used in identifying them.

| Technique Used | Bacteria Reported | Plant | Reference |
|---|---|---|---|
| Denaturing Gradient Gel Electrophoresis (DGGE) | *Sphingobacteriales, Flavobacteriaceae, Xanthomonadaceae, Cyanobacteria* | Lettuce, soybean, potato, maize | [35,36] |
| Quantitative PCR (qPCR) analysis | *Bacillus velezensis* NJAU-Z9 | Pepper | [37] |
| G3 PhyloChip microarray analyses | *Atribacteria, Dependentiae, TM6, Latescibacteria WS3 Marinimicrobia, SAR406; Omnitrophica, OP3; BRC1. Acidobacteria, Gemmatimonadetes,* and *Tenericutes* | Wheat, barley | [31] |
| Restriction Fragment Length Polymorphism (RFLP) | *Azospirillum, Pseudomonas chlororaphis, P. frederiksbergensis, Bacillus aryabhattai,* and *Paenibacillus peoriae* | Maize | [27] |
| DNA-Stable Isotope Probing (DNA-SIP) | *Nostocales, Stigonematales, Streptomyces Bacillus, Alicyclobacillus, Clostridium. Rhizobiales, Rhodospirillale, Myxococcales,* and *Actinomycetales* | Rice | [20,38] |
| 16S amplicon sequencing | *Proteobacteria, Actinobacteria, Bacteroidetes, Acidobacteria, Firmicutes, Verrucomicrobia, Planctomycetes, Actinobacteria, Cyanobacteria,* and *Gemmatimonadetes* | Wheat, maize, potato, soybean | [39,40] |
| Shotgun sequencing | *Stenotrophomonas, Rahnella, Sphingomonas, Janthinobacterium Luteibacter, Arthrobacter, Streptomyces, Bradyrhizobium, Methylobacterium, Ramlibacter, Nitrospira, Nocardioides, Geodermatophilus,* and *Burkholderia* | Soybean, sunflower, sugar beet | [26] |
| Culture-based | *Bacillus, Pseudomonas, Ochrobactrium, Providencia, Achromobacter, Burkholderia,* and *Enterobacter* | Wheat | [41] |

## 3. Biodegradation of Lignocellulose for Biofuel Production by Rhizospheric Bacteria

Agricultural and wood residues produced after harvesting and processing of plants are abundant biomass on earth, and these biomass resources have remarkable energy capacity [42]. Several tons (estimated to be around 10–50 billion produced annually worldwide [43]) of these wastes are produced annually from corn, wheat, soybean, timbers, etc., and are mainly composed of lignin, cellulose and hemicellulose. The lignocellulose biomass, recognized as the most abundant biopolymer on earth, is a predominant component of the cell wall of plants. It is an inexhaustible raw material for biofuel production. The plant cell wall is a heterogeneous complex of carbohydrate polymers (cellulose and hemicellulose) and an aromatic polymer (lignin) [44,45].

Lignin gives the plant a rigid structure that provides protection against the hydrolysis of cellulose and hemicellulose. In spite of the plethora of lignocellulose in nature, the expensive cost of hydrolyzing them into smaller monosaccharides has made the cost of using them unappealing. Therefore, this has led to the search for cheaper means of

hydrolyzing lignocellulose biologically [46]. Lignocellulose is used in many industrial processes for the production of chemicals, fuels, polymer precursors, paper and pulp, food and flavor compounds.

Presently, the search for and production of a renewable form of energy resources are of increased interest, which is the outcome of insubstantial areas for petroleum-based fuel production that is depleting and harming the environment [47]. For so many decades, plant materials and animal feed composed of carbohydrates have been used as biomass energy resources for substituting fuel production. Hence, the fuel produced from this form of resource is referred to as biofuels [43]. Several countries and the environment will gain from the profitable use of renewable fuel from biomass rather than petroleum-based automotive fuels [48]. The use of biofuel decreases the dependence on petroleum-based oil and the impacts of greenhouse gases, improves air quality and generates new employment opportunities [47]. The biofuel from lignocellulosic waste streams is ecofriendly, cost-effective and, thus, a priority worldwide. The main challenges of using this waste stream are the degree of polymerization, the protective lignin being recalcitrant to degradation and thus providing less surface area for enzymatic hydrolysis, the biomass particle size, and the crystalline nature of cellulose sheathed by hemicellulose [43].

Due to the recalcitrant nature of lignin toward degradation, the conversion of lignocellulose to biofuels and other renewable energy resources involves several pretreatment processes, such as biological, chemical, mechanical and thermal processes. The biological pretreatment process involves using microorganisms for the conversion and degradation of lignocellulose streams into sugars to produce biofuel [45]. The biological process is gaining popularity because it requires less energy, involves no chemicals and has less pollution.

Naturally occurring bacterial species from different niches have developed cellular mechanisms to acquire energy from plant biomass through the production and release of carbohydrate-active enzymes [47]. These enzymes degrade plant cell, which, as a result, synthesize monosaccharides that can be used biofuels and other value-added products. The use of plant biomass by bacteria is crucial for life on Earth because of their involvement in carbon flux in the environment [49]. The enzymes capable of degrading plant cell walls are widely used in industrial applications, including fuels and chemical production, and in the food and feed industry [47]. Enzymatic degradation of lignocellulose biopolymer is important for sustainable agricultural residues. These residues can be degraded to glucose by cellulases, such as cellobiohydrolase and endoglucanase [42].

The two main mechanisms used for biofuel production are direct and indirect fermentation processes [50]. The direct fermentation process is a microbial process that breaks down the starting plant materials into sugars capable of being fermented, and they are later transformed into alcohol [51]. Bacterial species, for example, *Clostridium, Lactobacillus, Microbacterium*, and *Leuconostoc* catalyze the production of fermentation products directly from the substrate [46,51]. This process does not require the step of converting the starting plant materials to gas. Comparatively, indirect fermentation uses the pyrolysis of the starting plant materials to generate a mixture of different gases, such as hydrogen, carbon dioxide, and carbon monoxide. Then, acetogenic microorganisms are used to transform the gases generated into ethanol [52].

The production of biofuel from lignocellulose requires different biological processes, such as delignification or pretreatment method (the release of free hemicellulose and cellulose from the lignocellulosic material), depolymerization (hydrolysis) of carbohydrate polymers from hemicellulose and cellulose to produce free sugars, and finally, the fermentation of mixed hexose and pentose sugars to produce ethanol [50]. Various pretreatment processes have been used to alter the plant cell wall, and these treatments are necessary to ensure the quality of products made from polysaccharides [53]. For example, in biofuel production, pretreatment is necessary because lignin prevents access of carbohydrate-degrading enzymes from binding, which limits product yield [45].

Biomass pretreatment methods are divided into physicochemical (steam pretreatment/auto hydrolysis, wet oxidation, and hydrothermolysis), physical (milling and grind-

ing), chemical (oxidizing agents, dilute acid, organic solvents, and alkali) and lastly, biological or a combination of the methods [54]. After that, the cellulose and hemicellulose are hydrolyzed into monomers by enzymatic or acid hydrolysis. Furthermore, for the fermentation process, bacteria are used to break down these monomeric sugars into alcohols [48]. The integration of these process configurations for biofuel production is efficient, cost-effective, and economical. The schematic process of biofuel production from lignocellulose biomass is depicted in Figure 1.

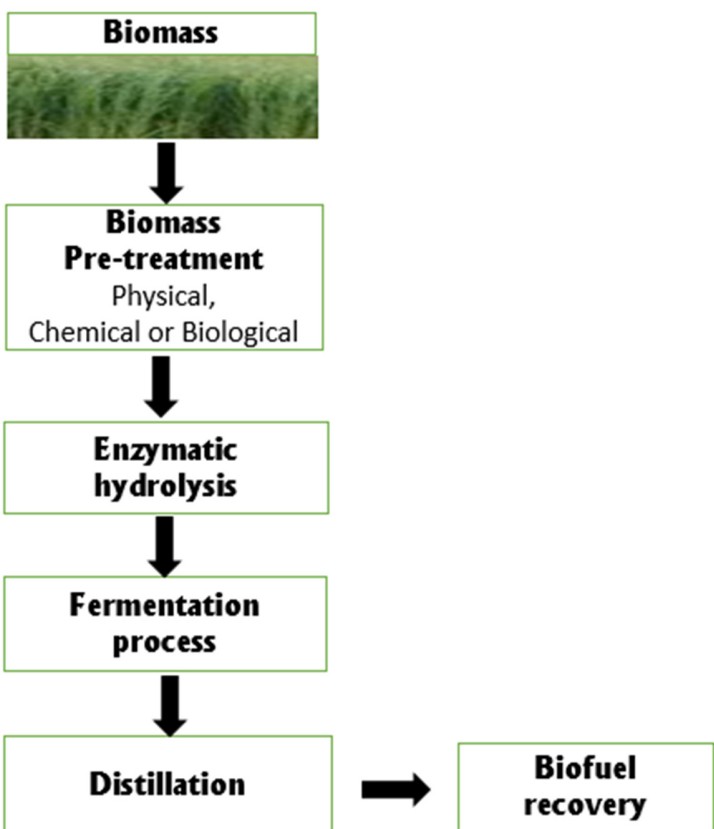

**Figure 1.** Schematic representation of the steps involved in biofuel production from lignocellulose biomass.

Lately, there is increased interest in rhizosphere soils because they house diverse bacterial populations that synthesize uncommon cellulolytic enzymes that are important in biofuel industries [55]. Cellulolytic enzymes such as monospecific endo-β-1,4-glucanase were obtained from rhizosphere soil through a metagenomics-based strategy for bioethanol production [56]. Cellulolytic bacteria such as *Erwinia*, *Sporocytophaga*, *Ruminococcus*, *Clostridium*, *Fibrobacter*, and *Cellulomonas* from the rhizosphere degrade cellulolytic materials containing small amounts of lignin [57].

Significantly, the degradation mechanism of lignocellulose by aerobic and anaerobic bacterial species differs due to its macromolecular arrangement [58]. Anaerobic bacteria often arrange the cellulase or hemicellulase apparatus into cellulosomes (multi-enzyme complex) composed of enzymes with different activities, including carbohydrate esterases, polysaccharide lysases, and glycoside hydrolases [59]. The catalytic constituents of cellulosomes are primarily made up of dockerins (a noncatalytic module's structure that coheres to cohesion modules) situated in the large non-catalytic protein acting as a stage. The association between cohesins and dockerins allows the incorporation of hydrolytic enzymes in the complex [60]. It has been reported that scaffoldins are in charge of binding the whole complex via a non-catalytic carbohydrate-binding module (CBM) to crystalline cellulose [59,60].

Many studies regarding cellulosomes are mainly focused on anaerobic bacteria such as *Clostridium* species [55]. Others include *Bacteroides cellulosolvens*, *Ruminococcus albus*, *Acetivibrio cellulolyticus*, and *Ruminococcus flavefaciens* [59,61]. The potential approach of advancing hydrolytic activity systems is the design and construction of cellulosome-based complexes [59]. Likewise, cellulosomes capable of incorporating bacterial enzymes from non-aggregating systems could be produced to augment hydrolytic activities and, accordingly, biomass saccharification [62].

Moreover, genetic manipulations are used to introduce genes capable of synthesizing cellulosome into bacteria that are able to degrade simple sugars [63]. However, they do not have functional machinery able to degrade plant cell walls. Alternatively, bacteria naturally synthesizing cellulosomes could be manipulated to increase their capacity to produce ethanol from lignocellulose [58]. Xylanosomes, which are self-organizing protein complexes, were designed and constructed using cellulosomes as template [64]. They were designed particularly for hemicellulose hydrolysis, but showed alliance with cellulases, and potential use of these nanostructures in cellulose hydrolysis has been recommended [59].

Rhizosphere soils are hotspots for bacterial phyla capable of degrading cellulose, such as *Acidobacteria*, *Firmicutes*, *Fibrobacteres*, *Alpha-proteobacteria*, *Actinobacteria*, and *Verrucomicrobia* [65,66]. Most strains of these phyla are degraders of plant biomass polysaccharides [67] and can also decompose lignin and phenolic compounds [46]. Examples of organisms isolated from the rhizosphere of various plants that produce cellulase and may be used in biofuel production are presented in Table 2. Many bacterial populations can transform complex lignocellulose polymers into monosaccharides with the aid of lignocellulolytic enzymes, which are essential for various far-reaching industrial processes, have been isolated from rhizosphere soils [68]. In a recent study, cellulose-degrading bacteria (*Bacillus*, *Chryseobacterium*, and *Pseudomonas*) were isolated from forage grass timothy (*Phleum pratense* L.) rhizosphere soil and endosphere [69]. Similarly, bacterial genes capable of degrading cellulose and xylan have been identified from bacteria such as *Acidobacteriaceae bacterium* from rhizosphere soil [52]. Table shows some rhizobacteria isolated from the rhizosphere of different plant species that can used in biofuel production.

**Table 2.** Rhizobacteria that produces enzyme that can be used in lignocellulose degradation for biofuel production.

| Rhizobacteria | Plant | Country | Reference |
|---|---|---|---|
| *Arthrobacter, Brevibacterium, Bacillus, Chryseobacterium, Stenotrophomonas, Streptomyces, Pseudomonas, Xanthomonas, Paenibacillus* | *Phleum pretense* L. | Canada | [69] |
| *Bacillus, Pseudomonas, Kocuria* | *Salsola stocksii* and *Atriplex amnicola* | Pakistan | [70] |
| *Streptomyces* | *Zea mays* | South Africa | [71] |
| *Arthrobacter, Pseudomonas* | *Quercus* sp. | Spain | [72] |

Mukhtar, et al. [70] reported thirty-eight and forty-five bacteria from the rhizosphere soil of *Atriplex amnicola* and *Salsola stocksii*, respectively. These organisms possessed xylanase, cellulase, and many other enzyme activities at 1.0–1.5 M NaCl concentration. The fermentation of simple sugars from lignocellulose degradation by these enzymes is an exceptional potential in biofuel production. Consequently, the knowledge of bacterial activities and environmental conditions influencing the alteration of vast quantities of carbon materials in rhizosphere soils could contribute to new opportunities that can benefit the environment (Table 3) [46,52,67,69].

**Table 3.** Biotechnological products manufactured from some rhizospheric bacterial phyla and its industrial applications

| Bacterial Phylum | Plant Rhizosphere | Biotechnological Product and Application | Industrial Application | Reference |
|---|---|---|---|---|
| Actinobacteria | *Helianthus annuus*, *Zea mays*, *Triticum aestivum*, *Glycine max* | Kanamycin enhanced shoot growth. | Actinobacteria secrete cellulases suitable for cellulosic biofuel production. | [73,74] |
| Proteobacteria | *Zea mays*, *Oryza sativa*, *Saccharum officinarum*, and *Glycine max* | Bioinoculants—significantly increased crop yield, biomass dry weight, nodulation, phosphorus, and nitrogen uptake. Bioprotectants—protect plant from phytopathogens. | Bioremediation strategies for the degradation of oil spill contamination (edible such as fats and lipids; and crude oil) as well as carbamate and organophosphate insecticides. *Zymomonas*—produces an abundance of alcohol for industrial use. Acetic acid bacteria can be employed for the production of acetic acid, vinegar, ascorbic (vitamin c), glucoronic, galactonic, arabonic acids, and sorbose. | [75] |
| Firmicutes | *Triticum aestivum* and *Vitis vinifera* | Nitrogen-fixing ability, enhance soil porosity and produce compound similar in activity to indole-3-acetic acid with the capacity to stimulate plant growth. Biocontrol activity effective against a wide range of phytopathogens, such as Fusaricidin identified as a potential antifungal agent, has been identified from *P. polymyxa* E68. In addition, control *Fusarium oxysporum*. Flocculants production. | *Paenibacillus polymyxa* produces 2, 3-butanediol (BDL) forms methyl ethyl ketone used as a liquid fuel additive by dehydration. Produce cell wall degrading enzymes (proteases, β-1,3-glucanases, xylanase, chitinases, and cellulases) available in detergent formulations, leather processing, food industry (starter culture for yogurt production, additives, and beer production), waste management and chemical synthesis. *Lactobacillus pentosus* has been applied in sulfite waste liquor fermentation. Flocculating or flocking agents used for water treatment. | [76] |

Table 3. *Cont.*

| Bacterial Phylum | Plant Rhizosphere | Biotechnological Product and Application | Industrial Application | Reference |
|---|---|---|---|---|
| Bacteroidetes | *Brassica napus* | Use alternative enzymatic mechanisms to solubilize biopolymers apart from glycosidic hydrolases, the so-called "polysaccharide-utilizer". Degrade complex polysaccharides in soils and contributes to synergistic breakdown of solubilized chitin oligosaccharides. | Produce enzymes exhibiting activities such as degrading cellulose, lignin or chitin. In addition, various lipids, polysaccharides, or proteins used in industries such as leather processing, detergent, paper, and shoe production Used for biofuel production. Used in phytoextraction of heavy metals from polluted soil. | [77,78] |
| Acidobacteria | *Castanea crenata, Saccharum officinarum, Vigna mungo* and *Solanum lycopersicum* L. | Produce exopolysaccharide (EPS), which provide protection against environmental stress and enable bacterial survival under unfavorable soil conditions. Form soil matrix, serve to sequester water and nutrition, and are involved in bacterial cell-surface adherence and soil aggregate formation. In addition, they produce plant growth-promoting traits and phytohormones. Produce biofilms, which enhance rhizobacterial root colonization by holding moisture and protect plant roots from phytopathogens. | EPS possess physical and chemical properties such as thickening, gelling, stabilizing, suspending, emulsifying, texture-enhancing, and coagulating. Although some of these bacterial products (e.g., gellan gum, dextran, alginate, and xanthan) have been commercialized successfully in the food and fodder production industries, EPSs are used as gelling, thickening, and suspending agents. For instance, xanthan (from *Xanthomonas campestris*) is used as a food additive. EPSs are bioemulsifiers that are used in the cosmetic and chemical (e.g., pesticide) industries. Use in environmental technologies, such as phytoremediation and bioremediation in soil and water by enhancing oil and heavy metal recovery. In addition, use in human health and chemical industries. | [79] |
| Nitrospirae | *Panax ginseng* Meyer | Plant growth promoter: Nitrite-oxidizing bacteria (NOB) are involved in nitrification, including the oxidation processes of ammonia and nitrite. In addition, they are known to convert nitrite to nitrate, improve shoot/root biomass, improve nutrient uptake, alleviate cold stress in plants, and serve as a biocontrol agent. | Potential in the petroleum industry for the exploration of petroleum, clean-up of oil spills both in situ and ex situ conditions and enhance microbial oil recovery. Bioconversion of food waste and activated sewage sludge into useful products. Biohydrometallurgy (microbial recovery of minerals from ores), used for fuel production and for clean-up of oil spills, and deterioration of petroleum products. | [80] |

**4. Biofertilization: The Use of Rhizosphere Bacteria as a Soil Amendment for Plant Growth Promotion**

Plant growth-promoting bacteria, such as *Azospirillum*, *Rhizobium*, and *Azotobacter* enhance the growth of different plants and, thus, are used for biofertilization of many crop plants [81]. The effects of plant growth stimulation have been attributed to atmospheric nitrogen fixation, potassium and phosphate solubilization, production of plant growth hormones (auxins, gibberellins, ethylene, and cytokinins), polyamines and diverse amino acids produced by PGPR, which improve the nutritional availability of plants directly [21]. There are several factors to be considered in the formulation of biofertilizer. These factors include the choice of appropriate microorganisms with the potential to colonize plant rhizosphere, the growth profile of the bacteria, appropriate carrier and types and optimum conditions of organisms. The success of the products also depends on the method of application and storage of the formulation [82].

Any material added to a soil to improve the physical and chemical properties of the soil, such as structure, water infiltration, water permeability, aeration, and drainage, which, as a result, provides a more suitable environment for plant roots and health, is referred to as soil amendment [32]. Owning to the distinct physical and chemical properties of rhizosphere soils due to microbial activities, they are the epicenter for nutrient accumulation. However, reports on soil and soil-like substrate amendments are limited. Currently, other methods widely employed as soil amendment include the excessive application of growth enhancers, chemical fertilizers, pesticides, and soil sterilization approaches, such as fumigation with methyl bromide (MeBr). Although these treatment methods can be efficient for controlling environmental stresses, they have harmful impacts on human health and the environment and long-term adverse effects on soil health and quality [83].

Promising alternatives such as using materials like bacterial soil amenders (PGPR) integrated with soil for enhancing the physical and chemical qualities of the amended soil, consequently improve plant health, performance, and growth. Plant growth-promoting rhizobacteria can improve crop growth through various mechanisms (direct or indirect), including improving water acquisition, increasing soil nutrient bioavailability, suppressing plant diseases, and decreasing herbivore damage [84]. Cui, et al. [85] inoculated maize with *Bacillus amyloliquefaciens* B9601-Y2, which consequently controlled the impacts of *Biopolaris maydis* (southern corn leaf blight) by colonizing the main roots and root hairs and later migrating to the stems and leaves. Likewise, *Bacillus amyloliquefaciens* B9601-Y2 notably improved maize-seedling growth (height, number of leaves), and chlorophyll content.

The physical and chemical parameters of rhizosphere and bulk soils are relatively variable [86]. This is due to different bacterial interactions and high nutrient concentrations. Several studies have revealed that the rhizosphere soils have significant levels of organic carbon, nitrogen, ammonium nitrate, clay content, and soil mineral nutrients such as phosphorus, iron, zinc, magnesium, copper, manganese, sodium, calcium, and potassium compared to bulk soils due to symbiotic nitrogen fixation, root exudation of organic acids, cluster root formation, plant secretion of phosphatases, and rhizosphere pH modification [86,87]. An experiment conducted by Maseko et al. [88] showed that the rhizosphere soils had approximately two-to-three times the phosphorus, copper, sodium, and potassium compared to the bulk soils. The clay content in rhizosphere soils increases their waterlogging and soil porosity. The high concentrations of these minerals naturally increase bacterial and phosphatase activities, which could be due to root exudation and, as a result, improve plant growth and yields [87]. Considering the exclusivity of the bacterial population in the rhizosphere soils, many authors have suggested their use as soil amendments in low-input farming systems [89] (Table 4).

**Table 4.** The rhizosphere soil and rhizobacteria as a soil amendment.

| Plant | Impact of Rhizobacteria on the Plant | Reference |
|---|---|---|
| *Capsicum annuum* L. | The soil amended with *Bacillus velezensis* improves seedling height, stem diameter, and yields compared to those pepper plants grown on un-amended soil | [37] |
| *Arabidopsis thaliana* | Combined mixture of rhizosphere soil or soil-like substrates and *Bacillus* mixtures resulted in a significant increase in plant root fresh weight, shoot fresh weight, nutrient uptake, chlorophyll content, and plant diameter. In addition, the transcript levels of ammonium and nitrate uptake genes in the plant were increased | [90] |
| *Helianthus annuus* | *Pseudomonas fluorescens* A506, *P. gessardii* strain BLP141, and *P. fluorescens* strain LMG 2189 improved plant growth, yield, physiology, proline, antioxidant activities, and reduced the malondialdehyde content in inoculated soil | [91] |
| *Ocimum basilicum* L. | Rhizobacteria consortium (*Bacillus lentus*, *Pseudomonas* sp. and *Azospirillum brasilens*) had positive effects on the antioxidant activity and chlorophyll pigment content under water-induced and salinity stress | [5] |
| *Festuca rubra* | Bacterial consortium immobilized in a mixture of perlite and sawdust (ratio 1:1:1 v/v) led to a substantial improvement of plant roots, stem length, and stem biomass, as well as influencing the elongation of the plants in all soil treated. Soil additives (phosphate fertilizer and sewage sludge) and an immobilized consortium of microorganism had a positive effect on plant growth (longer root, stem length, and stem biomass) compared to the control | [92] |
| *Eucalyptus globulus* | Co-application of biochar ($20$ t hm$^{-2}$) and PGPB ($5 \times 10^{10}$ CFU mL$^{-1}$) amendments significantly decreased the concentrations of soil total P and NH$_4^+$-N, whereas they advanced total K, NO$_3$-N, and soil water content, and hence maintained soil sustainability in eucalyptus plantation | [93] |
| *Curcuma longa* | *The Curcuma longa* soil amended with B. subtilis MML2490 and *P. aeruginosa* MML2424 enhanced plant growth promotion and management of turmeric rhizome rot disease, and thus appeared promising for commercialization | [94] |

A major challenge with applying and using biofertilizers on the field is their inconsistency and unreliability, which calls for innovative solutions. This could be as a result of the diverse growth habitat and community structure of plant roots. This can create a stressful environment for the proliferation of the product [95]. Another limiting factor for their use is their selectivity, resulting in variable quality and efficiency on the field. Innovative solutions that direct the product to the desired location and target crop and an understanding of the root microbiome dynamics and flux in plant metabolic networks can increase the chances of product effectiveness and reproducibility [96].

## 5. The Role of Rhizosphere Soil and Its Bacteria for Bioremediation and Biofiltration

Recently, researchers have given in-depth attention to bioremediation, which is a process that mainly stimulates and uses microorganisms such as bacteria, plant enzymes, or plants to degrade and treat target pollutants in soil and other environments by manipulating the environmental conditions [91]. Phytoremediation, a combination strategy used by plants and microorganisms to remove pollutants from the environment, has proven to be effective in decontaminating soil polluted with heavy metals and hydrocarbons. Some bacterial species have synergistic interactions (direct or indirect) with environmental factors capable of removing hydrocarbon and heavy metal pollutants from soils, which consequently successfully increases the movement of these pollutants to the above-ground plants' biomass. Rhizoremediation, a phytoremediation approach, is plant–microbe cooperation with the potential to remove soil pollutants through the action of microorganisms and plant enzymes in the rhizosphere [97].

Many PGPR strains capable of removing pollutants from environmental media are found in the rhizosphere soils [98]. dos Santos and Maranho [33] reported the efficiency of *Bacillus*, *Alcaligenes*, *Microbacterium*, and *Curtobacterium* isolated from rhizosphere soil in heavy metal transformation. *Pseudomonas fluorescens* isolated from rhizosphere soil was inoculated into maize soil polluted with cadmium in research conducted by Asilian,

et al. [99]. Their discovery revealed that maize soil amended with 2 mmol kg$^{-1}$ Tween 80 had a better Cd uptake than the control (soil uninoculated). Moreover, they reported that soil samples inoculated with 4 mmol kg$^{-1}$ Tween 80 increased translocation efficiencies and phytoextraction (a subprocess of phytoremediation in which plants remove heavy metals from soil or water even at relatively low concentrations) than soil uninoculated with *P. fluorescens.* Therefore, they proposed that phytoextraction of heavy metals can be achieved by *P. fluorescens.* The properties and mineral compositions of rhizosphere soil place them as an excellent accumulator (adsorbent) of heavy metals and a promising technique for soil remediation capable of purifying soil from contaminants [84].

This is evident in research where Mourato, et al. [100] used lead (Pb) metal phytoextraction to reveal that *Brassica nigra* and *Brassica juncea* rhizosphere soils have higher heavy metal-bioaccumulating ability than those that are uninoculated. Other salient findings showed that cultivar of *B. juncea* rhizosphere soil was the most adsorbent of Pb (up to 3.5% on a dry weight basis) compared to the uninoculated soil [101]. Hence, the tight binding of Pb to plant and soil materials explains the low movements in soil and plants. In another experiment, the researchers reported using grass species (*Cenchrus ciliaris*) irhizosphere soil as a tool for removing Pb and Cu from aqueous solutions [102]. The result from this investigation revealed that the rhizosphere soil absorbed Pb up to 97.31 ppm and 188.3 ppm Cu. They concluded that the rhizosphere remediation using *C. ciliaris* rhizosphere is a compelling and effective green innovation for remediation of heavy metals from soil. Xu, et al. [103] used cadmium (Cd$^{2+}$) resistant *Pseudomonas* sp. strain 375 from heavy metal polluted rhizosphere soil as an adsorbent to remediate water body polluted with Cd$^{2+}$. *Pseudomonas* sp. was used as an inexpensive and potential bioadsorbent for bioremediation of Cd$^{2+}$ from wastewater. Other rhizobacteria and pollutants they removed are listed in Table 5.

Additionally, Canizo, et al. [104] stated that bacterial byproducts such as Extracellular polysaccharides or Exopolysaccharides (EPS) can be employed for bioremediation of heavy metals from soil and water. The review gave an extensive assessment of the biosorption of crystal violet (CV) dyes from effluents and natural water using *Rhodococcus erythropolis* AW3 biomass as a biosorbent. The results revealed that Langmuir isotherm model had the highest biosorption capacity (289.8 mg g$^{-1}$) in removing CV dye from effluents and natural water by *R, erythropolis*.

Through the process of decomposition, bacterial species can decompose plant materials to produce a remarkable number of greenhouse gases. Yuan, et al. [105] reported that paddy soil adds 10% of total global atmospheric methane (CH$_4$) emissions. However, this depends on the bacterial species involved. Nevertheless, rhizosphere soils harbor many methane-oxidizing bacteria such as acetoclastic *Methanosaeta* and hydrogenotrophic *Methanocella* that function as biofilters. These bacteria can adsorb the methane produced in the soil by bacteria and reduce the quantity of methane released into the surroundings.

In a study to ascertain if bacteria in the rhizosphere soil can mitigate the emission of methane, Aimen, et al. [106] reported that methanotrophic bacteria colonizing the rhizosphere soil were capable of oxidizing most of the methane formed in the rhizosphere before it was discharged. This is because *pmoA* and *mcrA* genes, which encode methane monooxygenase subunit, possess a strong affinity for methane; even at a low concentration of 6.25 kg ha$^{-1}$ methanotrophs can minimize the emission of methane by 60% [107].

**Table 5.** Rhizobacteria and pollutants removed from the environment.

| Rhizobacteria | Isolation Source | Pollutant | Reference |
|---|---|---|---|
| *Bacillus thurigiensis, B. pumilus* and *Rhodococcus hoagii* | *Panicum aquaticum* | Petroluem | [108] |
| *Lysinbacilus fusiformis* L8, *Bacillus weihenstephanensis* UT11, *Paenibacillus* sp. M10-6, | *Hosta undulata* | Alkylphenol | [109] |
| *Ensifer, Novosphingobium, Norcardioides, Streptomyces, Rhizobium* | *Coronilla varia, Vigna unguiculata* | Phenantrene | [110] |
| *Mycobacterium gilvum* | *Phragmites australis* | Pyrene, benzo[a]pyrene | [111] |
| *Bacillus subtilis, Bacillus amyloliquefaciens* | *Lactuca sativa* L. | Cadmium | [112] |
| *Microbacterium hydrocarbonoxydans, Achromobacter xylosoxidans, Bacillus subtili, B. megaterium, Alcaligens faecalis, Pseudomonas migulae* | *Phragmites australis* | Colored distillery effluent | [113] |
| *Alcaligenes, Bacillus, Curtobacterium, Microbacterium* | *Prosopis laevigata, Spharealce aangustifolia* | As(V), Pb(II), Cu(II), Zn(II) | [114] |
| *Bacillussp.* CIK-512 | *Zea mays* | Pb | [115] |

## 6. Concluding Remarks and Future Perspective

Biotechnology is an important and rapidly developing field of technology worldwide for its significant contribution to food, health, and environmental sustainability. In the quest of employing naturally occurring materials for biotechnological purposes, we reviewed rhizosphere soil bacterial materials as a prospective essential tool. This is because they house important beneficial bacteria useful in the production of biofuel, bioremediation of heavy metals, and biofilteration of gases, and can also be used as soil amendment. To maximize the complete benefits of rhizosphere soil for biotechnological purposes, there is a need for further critical research to use the metabolic potentiality of bacteria found in the rhizosphere soils, subsequently unveiling their complete potential. Moreover, to certify a sustainable application of rhizosphere bacterial materials in the future, experiments should be performed to ameliorate factors that stimulate and improve rhizosphere soil restoration by bacteria.

**Author Contributions:** B.C.N., managed the literature searches and wrote the first draft of the manuscript. A.S.A., provided academic input, managed the literature search and thoroughly critiqued the manuscript. O.O.B. proof-read the drafts, critically reviewed various drafts and provided scientific imputs. All authors have read and agreed to the published version of the manuscript.

**Funding:** National Research Foundation, South Africa, funded the work in our laboratory (UID123634).

**Institutional Review Board Statement:** The study did not require ethical approval.

**Informed Consent Statement:** Not applicable.

**Data Availability Statement:** Not applicable.

**Acknowledgments:** BCN thanks the National Research Foundation South Africa/The World Academy of Science (NRF-TWAS) for the Ph.D. stipend (UID121772). OOB acknowledges the National Research Foundation, South Africa for a grant (UID123634) that supported research in her laboratory. ASA appreciates North-West University for the postdoctoral fellowship granted to him.

**Conflicts of Interest:** No potential conflict of interest was reported by the authors.

**Ethical Approval:** This article does not contain any studies with human participants or animals performed by any of the authors.

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
