# Peer review of "Elucidating the Rhizosphere Associated Bacteria for Environmental Sustainability"

_agriculture, doi:10.3390/agriculture11010075_

Round 1

Reviewer 1 Report

This review manuscript has attempted to review the rhizospheric bacteria for environmental sustainability. However, there is no any novel information in this review. And the author have not followed any systematic pattern to review the literature of rhizospheric bacteria for environmental sustainability. There are several flaws prevailed in this manuscript.

Comments:

  1. Introduction: The two third part of the manuscript is not well illustrated focusing on the main theme of this review. This section lacks the brief literatures on the recent explanation of the recent trend of rhizospheric bacteria for environmental sustainability. A rational to conduct this review is lacking from the introduction.
  2. In Table 1 the author have highlighted the identification technique. But, there is no any description focused on this topics in section 2. Why the author has highlighted “Technique used” in the table 1? Please clarify on section 2. In addition, please add some more information on root exudates that support the bacterial growth. Please provide what specific roost exudates help to proliferate the bacterial population in the roots. On the other hand the author can include about the mutualistic mechanism of bacteria and roots. A figure more effective to illustrate these information.
  3. Section 3: Please provide the brief information in tabulated form on the rhizospheric bacterial species involved in biofuel production. Also, it is recommended to provide biofuel production mechanism by bacteria by making schematic figure.
  4. Section 4:What are the basic factors to be considered in applying bacteria as biofertilizer? What are the recent trend in developing bioferilizers? What is the survivability of bacteria after applying in the crop filed? These points can be valuable in this section.
  5. Please provide a table illustrating the various rhizospheric bacteria involved in the bioremediation field. There are several recent papers where researcher have employed and study the synergistic effect of bacteria in applying as bioremediation tools.
  6. There is no any future prospective mentioned in this review.

Author Response

This review manuscript has attempted to review the rhizospheric bacteria for environmental sustainability. However, there is no any novel information in this review. And the author have not followed any systematic pattern to review the literature of rhizospheric bacteria for environmental sustainability. There are several flaws prevailed in this manuscript.

Comments:

  1. Introduction: The two third part of the manuscript is not well illustrated focusing on the main theme of this review. This section lacks the brief literatures on the recent explanation of the recent trend of rhizospheric bacteria for environmental sustainability. A rational to conduct this review is lacking from the introduction.

Response: Recent trend has been included and some redundant part of the introduction remove. The rationale for the review has been included in the last paragraph

Environmental sustainability acknowledges the importance to advance and control the biological and physical systems that bolster both the short- and long-term value of all forms of life on earth without jeopardizing the diversity and well-being of natural ecosystems [13]. By virtue of the ecological services rendered by rhizobacteria, Ambrosini, et al. [14] have recommended further research on the factors that aid in the maintenance of the rhizosphere bacterial community and promote practices that advance rhizosphere conservation and protection. Despite the critical role played by rhizobacteria in redressing soil fertility and environmental sustainability, there still remains the need for further understanding of the mechanisms through which rhizobacteria perform their ecological roles and how such roles can be exploited for environmental sustainability. Therefore, critical discussion on the diversity of bacteria in the rhizosphere soils and their role in lignocellulose degradation, biofuel production, biofiltration, and bioremediation, as well as the possibility of achieving soil amendment were provided.” (line 62-72).

  1. In Table 1 the author have highlighted the identification technique. But, there is no any description focused on this topics in section 2. Why the author has highlighted “Technique used” in the table 1? Please clarify on section 2. In addition, please add some more information on root exudates that support the bacterial growth. Please provide what specific roost exudates help to proliferate the bacterial population in the roots. On the other hand the author can include about the mutualistic mechanism of bacteria and roots. A figure more effective to illustrate these information.

Response: The description of the technique has been included in line 134-136.

“Despite variations in the dynamics and composition of root exudates, a subset of the bacterial population are designated as the core rhizosphere microbiome, which are ubiquitous across plant species and environment [17]. The microbiota uses the root exudates as a source of energy, and the common genera in the rhizosphere include Burkholderia, Bacillus, Microbacterium, Azospirillum, Serratia, Pseudomonas, Erwinia, Aeromonas, Mesorhizobium, Rahnella, Acinetobacter, Enterobacter and Acinetobacter [31,32]. Conventionally, bacterial species in the rhizosphere were isolated and identified using the traditional or culture-based method for isolating and classifying microorganisms. This method's main inadequacy is that it cannot identify the entire microorganism in a sample, making approximately 99% of the microorganism unknown [6]. Thus, only a few bacterial population has been identified from the rhizosphere soils using conventional techniques, but not until the advent of next generation sequencing techniques [33]. High-throughput sequencing has made the identification of most rhizobacteria possible and also lend credence to their functional role in the rhizosphere (Table 1).’

More information about the root exudate and how they help in the proliferation of rhizobacteria has been included in line 108-124

‘Root exudation includes the secretion of enzymes, oxygen and water, ion, mucilage and diverse carbon-containing metabolites. The plant root system produces various metabolites, while the root tips secrete most of the root exudates, which are low molecular weight organic substances (such as amino acids, amides, organic acids, sugars, enzymes, phenolic acids, coumarin, high-molecular weight compounds (such as proteins and mucilages) and other substances, including sterols that attract bacteria to the rhizosphere [26,27]. However, the components of the exudates vary in the amount released, molecular weight, and biochemical functions. These exudates act as attraction signals that influence the ability of bacteria to colonize the roots. To proliferate and be established in the rhizosphere, the organisms must be able to use root exudates, colonize the root or rhizosphere effectively and be able to compete with other organisms [23]. Rhizobacteria locate plant roots through cues exuded from the roots and root exudates, which stimulate PGPR chemotaxis on root surfaces. Root exudates can also stimulate flagella motility in some rhizobacteria [28]. These traits are essential for colonization of the rhizosphere.

The impact of plant roots was examined on rhizosphere and bacterial communities, and it was deduced that root length, biomass, density, volume, and surface area create distinct ecological niches for some bacterial species to improve advantageous interactions in the rhizosphere [29]. It has been established that since the root tips make initial connection with the bulk soil, the bacterial communities and rhizodeposits are notable in maintaining the rhizosphere [4,30].’

  1. Section 3: Please provide the brief information in tabulated form on the rhizospheric bacterial species involved in biofuel production. Also, it is recommended to provide biofuel production mechanism by bacteria by making schematic figure.

Response:

Rhizobacteria species involved in biofuel production has been included in Table 2. (line 261). Similarly, the process of biofuel production has been provided in figure 1. (line 226).

  1. Section 4: What are the basic factors to be considered in applying bacteria as biofertilizer? What are the recent trend in developing bioferilizers? What is the survivability of bacteria after applying in the crop filed? These points can be valuable in this section.

Response:

The factors to be considered in applying biofertilizers on the field has been included in line 269-272.

‘There are several factors to be considered in the formulation of biofertilizer. These factors include the choice of appropriate microorganisms with the potential to colonize plant rhizosphere, the growth profile of the bacteria, appropriate carrier and types and optimum conditions of organisms. The success of the products also depend on the method of application and storage of the formulation’.

Their survival and the recent trend has been included in line 314-320

“A major challenge with the application and use of biofertilizers on the field is their inconsistency and unreliability, which calls for innovative solutions. This could be as a result of the diverse growth habitat and community structure of plant roots. This can create a stressful environment for the proliferation of the product [90]. Another limiting factor for their use is their selectivity, which can result in variable quality and efficiency on the field. Innovative solutions that directs the product to the desired location and target crop as well as understanding of the root microbiome dynamics and flux in plant metabolic networks can increase the chances of product effectiveness and reproducibility”

  1. Please provide a table illustrating the various rhizospheric bacteria involved in the bioremediation field. There are several recent papers where researcher have employed and study the synergistic effect of bacteria in applying as bioremediation tools.

Response:

The rhizobacteria involved in the degradation of various pollutants in the environment has been provided in Table 5 (line 381).

  1. There is no any future prospective mentioned in this review.

Response:

The future prospect has been provided in the concluding section (line 389-394)

“To maximize the complete benefits of rhizosphere soil for biotechnological purposes, there is a need for further critical research to use the metabolic potentiality of bacteria found in the rhizosphere soils, subsequently unveiling their complete potential. More so, to certify a sustainable application of rhizosphere bacterial materials in the future, experiments should be performed to ameliorate factors that stimulate and improve rhizosphere soil restoration by bacteria. “

Reviewer 2 Report

The manuscript is concerning the Rhizosphere associatec bacteria in the perspective of an environmental friendly plant development.

In my opinion, such review is well written and all the chapter are sounding good to me. The writting level is quite good but there is still a few typos or non sense sentence to me.

Line 68: I do not understand. In my opinion, this sentence should be rephrase !

Line 115: I do not see any link between the first sentence and the last sentence.

Line 133: Authors sould consider to rewrite this sentence.

Line 179: two "dot" at the end of the sentence.

Line 200: I suppose that "in" should be replaced by "is"

Line 220: Correct "as a stage" to "as that stage"

Line 263 and at line 316: Authors should use PGPB acronym as described at line 43.

Line 339: Add sp. "strain" 375, it is more easy to understand.

Line 343: Extra bracket at the end of (EPS))

Line 347: Replace "," by "." between R. erythropolis.

Line 103: The review should be written " Water Air Soil Poll." without extra dot.

Author Response

The manuscript is concerning the Rhizosphere associatec bacteria in the perspective of an environmental friendly plant development.

In my opinion, such review is well written and all the chapter are sounding good to me. The writting level is quite good but there is still a few typos or non sense sentence to me.

Comment: Line 68: I do not understand. In my opinion, this sentence should be rephrase !

Response: The sentence has been removed and the paragraph rephrased

Environmental sustainability acknowledges the importance to advance and control the biological and physical systems that bolster both the short- and long-term value of all forms of life on earth without jeopardizing the diversity and well-being of natural ecosystems [13]. By virtue of the ecological services rendered by rhizobacteria, Ambrosini, et al. [14] have recommended further research on the factors that aid in the maintenance of the rhizosphere bacterial community and promote practices that advance rhizosphere conservation and protection. Despite the critical role played by rhizobacteria in redressing soil fertility and environmental sustainability, there still remains the need for further understanding of the mechanisms through which rhizobacteria perform their ecological roles and how such roles can be exploited for environmental sustainability. Therefore, critical discussion on the diversity of bacteria in the rhizosphere soils and their role in lignocellulose degradation, biofuel production, biofiltration, and bioremediation, as well as the possibility of achieving soil amendment were provided.

Comment: Line 115: I do not see any link between the first sentence and the last sentence.

Response: The statement has been linked together

“However, shaping and establishment of the rhizosphere microbiome is a selective and dynamic process that involves several mechanisms such as signal recognition, chemotaxis, biofilm formation and antibiosis”. (line 105-107)

Comment: Line 133: Authors sould consider to rewrite this sentence.

Response: The statement has been rephrased thus “It has been established that since the root tips make initial connection with the bulk soil, the bacterial communities and rhizodeposits are notable in maintaining the rhizosphere” (line 122-124)

Comment: Line 179: two "dot" at the end of the sentence.

Response: The extra dot has been removed

Comment: Line 200: I suppose that "in" should be replaced by "is"

Response: It has been replaced (line 201)

Comment: Line 220: Correct "as a stage" to "as that stage"

Response: It has been corrected (Line 222)

Comment: Line 263 and at line 316: Authors should use PGPB acronym as described at line 43.

Response: The acronym has been used (Line 266 and 333)

Comment: Line 339: Add sp. "strain" 375, it is more easy to understand.

Response: Strain 375 has been included in the name of the organism (Line356)

Comment: Line 343: Extra bracket at the end of (EPS))

Response: The extra bracket has been removed

Comment: Line 347: Replace "," by "." between R. erythropolis.

Response: The word has been replaced by “by” (Line 365)

Comment: Line 103: The review should be written " Water Air Soil Poll." without extra dot.

Response: The journal name has been changed to “Water Air Soil Poll.” (Line 707)

Round 2

Reviewer 1 Report

The manuscript has been revised well.